# Partial Substitution of Glucose with Xylitol Prolongs Survival and Suppresses Cell Proliferation and Glycolysis of Mice Bearing Orthotopic Xenograft of Oral Cancer

**DOI:** 10.3390/nu14102023

**Published:** 2022-05-11

**Authors:** Yuraporn Sahasakul, Wannee Angkhasirisap, Aroonwan Lam-ubol, Amornrat Aursalung, Daisuke Sano, Kentaro Takada, Dunyaporn Trachootham

**Affiliations:** 1Institute of Nutrition, Mahidol University, Salaya, Phuttamonthon, Nakhon Pathom 73170, Thailand; yuraporn.sah@mahiol.ac.th (Y.S.); amornrat.aur@mahidol.ac.th (A.A.); 2National Laboratory Animal Center, Mahidol University, Phuttamonthon, Nakhon Pathom 73170, Thailand; wannee.ang@mahidol.edu; 3Department of Oral Surgery and Oral Medicine, Faculty of Dentistry, Srinakharinwirot University, Bangkok 10110, Thailand; aroonwan@gmail.com; 4Department of Otorhinolaryngology-Head and Neck Surgery, School of Medicine, Yokohama City University, 3-9 Fukuura, Kanazawa-ku, Yokohama 236-0004, Japan; dsano@yokohama-cu.ac.jp (D.S.); drktakada@yahoo.co.jp (K.T.)

**Keywords:** oral cancer, orthotopic xenograft model, xylitol, glucose, glycolysis, sweetener, PFK

## Abstract

Many types of cancer have metabolic alterations with increased glycolysis. Identification of alternative sweeteners that do not fuel cancer is a novel approach to cancer control. The present study aimed to investigate the effects of xylitol on tumor growth and survival of mice bearing orthotopic xenograft of tongue cancers. The results showed that partial substitution of glucose with xylitol (glucose 0.35 g plus xylitol 2.06 g/kg body weight) non-significantly reduced tumor volume, and significantly prolonged the median survival time from 19 days in the control to 30.5 days in the xylitol group. Immunohistochemical data of the tongue tissue shows significantly lower intense-to-mild staining ratios of the proliferation marker Ki-67 in the xylitol than those of the control group (*p* = 0.04). Furthermore, the xylitol substitution significantly reduced the expression of the rate-limiting glycolytic enzyme, phosphofructokinase-1 (PFK-1) (*p* = 0.03), and showed a non-significant inhibition of PFK activity. In summary, partial substitution of glucose with xylitol at the equivalent dose to human household use of 10 g/day slows down tumor proliferation and prolongs survival of mice bearing an orthotopic oral cancer xenograft, possibly through glycolytic inhibition, with minimal adverse events. The insight warrants clinical studies to confirm xylitol as a candidate sweetener in food products for cancer survivors.

## 1. Introduction

Cancer is a leading cause of death worldwide [1,2]. According to the GLOBOCAN report, an estimated 19.3 million new cancer cases occurred in 2020, and of this number, oral cancer accounts for 377,713 cases and 177,757 deaths [2]. The highest incidence of oral cancer is found in Asia [3]. From 1990 to 2017, the global incidence and mortality rate of oral cancer have doubled [4]. Importantly, oral cancer survivors are at high risk for developing cancer recurrence or secondary cancer 5–10 years after the first diagnosis [5,6]. Survival of patients is influenced by disease progression and lifestyle-related behaviors, especially dietary consumption. As a result, dietary selection plays an important role in enhancing cancer survivorship and preventing recurrence [6,7]. Adherence to the Mediterranean diet lifestyle was associated with improved survival of cancer patients [8,9]. Besides consuming more legumes, vegetables, fruits, nuts, wholegrain foods, and fish, the Mediterranean diet pattern also promotes consuming fewer sweets made from sugar [10].

Glucose is the primary energy source for all cells and tissues. In normal cells under normal oxygen conditions, glucose will be converted to pyruvate via glycolysis. Then, it will be changed to Acetyl Co-A before entering the mitochondria to undergo a tricarboxylic acid cycle (TCA cycle), followed by an electron transport chain, which requires oxygen. This aerobic process can generate 36 ATP per glucose [11]. Under the hypoxic condition, normal cells will use anaerobic glycolysis and pyruvate will be converted to lactate, yielding 2 ATP/glucose [11]. In contrast, cancer cells, regardless of oxygen condition, have increased glucose uptake, and prefer to use glycolysis with no need for oxygen to generate 4 ATP/glucose. This phenomenon is called the Warburg effect [11,12]. This metabolic alteration toward the fast and efficient ATP generation by glycolysis can support rapid cancer cell proliferation and progression [12,13]. The most important rate-limiting enzyme of glycolysis is phosphofructokinase (PFK), and upregulation of this enzyme is found in many types of cancer, suggesting the altered metabolism [13]. Owing to the importance of this metabolic shift on cell growth, the inhibition of glycolysis using small molecules to target glycolytic enzymes is an emerging approach for cancer therapy [12,14,15].

Since many types of cancer cells, except prostate cancer, are addicted to glucose, it is crucial to identify an alternative source of carbohydrate that normal cells can utilize but that does not promote the proliferation of cancer cells [7,12]. Naturally occurring sugar alcohols such as xylitol are a potential candidate due to their different energy metabolism than that of glucose [7,16,17]. Xylitol is widely used as a sweetener in confectioneries and gums [16,18,19]. It has only one-third of the caloric content compared to that of conventional sugar (2.4 Kcal/g) [18,20]. Its sweetness is equal to that of sucrose [16]. It has anti-cariogenic properties [21,22], anti-inflammatory effects [23,24], and was recently found to inhibit lung cancer cell proliferation [23] and induce apoptosis in melanoma cells [25,26]. Interestingly, our previous in vitro studies demonstrated that partial substitution of glucose in culture media with xylitol significantly suppressed the proliferation of oral cancer cells. Promisingly, the proliferation of non-transformed oral keratinocytes was not affected by the xylitol treatment. Despite the addition of iso-caloric quantities of the sugars, cancer cells exposed to partial substitution of glucose with xylitol had more retarded ATP generation, compared to those receiving the glucose control. Mechanistic studies suggest that the selective anti-proliferative effect of xylitol likely results from the inhibition of glucose utilization by suppressing the rate-limiting glycolytic enzyme PFK [7].

Since the in vitro effect of xylitol is promising, this study was intended to explore the in vivo effect of xylitol against oral cancer. Currently, the animal model that most resembles human head and neck cancer is the orthotopic tongue xenograft model, which was developed by Sano and colleagues [27,28]. In this orthotopic model, the human tongue cancer cells have been inoculated into the tongue of animals; thus, tongue cancer was generated at a similar location as its origin. Human tongue cancer cell line CAL-27 has been successfully used in a xenograft model with a 100% tumor formation rate [29]. Thus, the CAL-27 orthotopic model was used here. For the treatment with xylitol, we compared the control group receiving the glucose solution to that of partial substitution of glucose with xylitol to achieve equal energy input. Such design is intended to imitate the possible realistic use of xylitol as a partial sugar substitute. The dose of xylitol used in this work was converted from the average household use of xylitol in humans at 5–10 g/day (0.1–0.2 g/kg body weight) to the animal equivalent dose. Therefore, the goals of this study were to investigate the effect of partial substitution of glucose with xylitol at the human equivalent dose of 10 g/day on tumor volume, survival time, and cell proliferation, and to elucidate the potential mechanism using the orthotopic tongue xenograft model. Since the mice bearing cancer in their tongue would have trouble eating, in this study we also provided a texture-modified diet and hydrogel water to resemble what has been implemented for nutritional support of cancer patients.

## 2. Materials and Methods

### 2.1. Sample Size and Power Calculation

The primary outcome of this study was the comparison between the average tumor volume of experimental groups 1 and 2 and the control. Therefore, we calculated the sample size using G*Power V.3.1.9.2 software based on one-way ANOVA using the following parameters: effect size = 0.8, α = 0.05, power = 0.95, and number of groups = 3.

The effect size from the mean was determined based on two previous studies, i.e., the estimated standard deviation within the group was from a study by Sano et al. of an orthotopic xenograft model using the CAL-27 cancer cell line [29]. The estimated means of the three groups were from an intervention study by Oh et al. using the orthotopic xenograft model [30]. The software calculated and yielded the following result: non-centrality parameter λ = 19.2, critical F = 3.35, numerator df = 2, denominator df = 27, and total sample size = 30, or 10 each for 3 groups. Based on our previous experience, after tumor inoculation, not all mice develop a similar baseline volume of tumor at a similar time. Therefore, five more mice were initially added to ensure that in the end, we would have at least thirty homogeneous mice for grouping and intervention.

After completing the experiments, the result showed a significant difference in the survival of mice in the high dose of the xylitol group compared with that of the control group. To ensure that the sample size was adequate for the statistical analysis of this data, a post hoc power calculation for the log-rank test was performed by using the online calculator at the website: https://homepage.univie.ac.at/robin.ristl/samplesize.php?test=logrank (accessed on 14 April 2022). Using the hazard ratio of 4.111, alpha 0.05, and the sample size of 20, the post hoc power was 0.8851, which is considered adequate (≥0.8).

### 2.2. Animals and Maintenance

Male athymic BALB/c Mlac-nu mice, age 8 weeks old, were purchased from M-CLEA Bioresources (Samutprakarn, Thailand). The mice were housed and maintained in a laminar flow cabinet under specific pathogen-free conditions, with a 12 h light/dark cycle, and temperature was controlled at 22 ± 3 °C. The animal use was performed according to the principles proposed by the National Centre for the Replacement, Refinement, and Reduction of Animals in Research (NC3Rs). The animal protocol of this study was pre-approved by the Institutional Animal Care and Use Committee (IACUC) of the National Laboratory Animal Center (NLAC), Mahidol University, Thailand (NLAC-MU Protocol No. RA 2015-09). During the acclimatization period, mice were fed a sterile regular diet (Standard diet 086, Charoen Pokphand) and bottled water. After tumor initiation on the tongue, the mice had difficulty consuming the pellet diet and bottled water. Thus, a texture-modified diet (diet to water ratio of 1:1.5) was prepared, sterilized, and fed to all animals ad libitum. It was found to improve the diet consumption of tumor-bearing animals. Apart from bottled water, hydrogel (1% agar in water) was also supplied as a good alternative source of water for tumor-bearing mice. Body weight was monitored every week. The amount of food, water, and hydrogel intake was monitored every day.

### 2.3. Orthotopic Tongue Cancer Xenograft Model

CAL-27 tongue carcinoma cells were obtained from American Type Culture Collection (ATCC). Dulbecco’s modified Eagle’s medium (DMEM)-F12 and ES cell-qualified fetal bovine serum (FBS) were from Gibco, Thermo-Fisher Scientific, Waltham, MA, USA. Cell culture was conducted at Cell Biology Laboratory, Institute of Nutrition, Mahidol University. CAL-27 cells were grown in DMEM-F12 with 10% Fetal Bovine Serum at 37 °C in a 5% CO_2_ incubator. After being harvested by trypsinization, CAL-27 cells were suspended in serum-free media, and placed on ice for carrying to the animal house until the inoculation. At NLAC, all of the mice were anesthetized with isoflurane for stabilization. Then, tumor cells were inoculated as in previous studies [27,29]. In brief, 30 µL of serum-free media containing 5 × 10^4^ CAL-27 cells was injected submucosally into the lateral tongue of each animal, using a 0.5 mL insulin syringe (BD Medical—Diabetes Care, Holdrege, Nebraska, NE, USA) with a 30G × 1/2 (0.3 × 13 mm) PrecisionGlide ^TM^ needle (Becton, Dickinson and Company, Franklin Lakes, NJ, USA).

For the full-scale experiment, 35 mice were inoculated with CAL-27 cells to establish an orthotopic xenograft model. However, only 30 mice had developed measurable tumors and were used for the interventions.

### 2.4. Interventions

Glucose powder was obtained from the Pharmacy department, Siriraj Hospital (Bangkok, Thailand). Food-grade XIVIA^®^ Xylitol was a product of Danisco (London, UK). Thirty animals with measurable tumors were randomly and evenly allocated into three groups. As shown in Table 1, all groups received equal energy input of the following: (1) control: glucose 1.59 g/kg body weight, (2) glucose 0.97 plus xylitol 1.03 g/kg body weight, and (3) glucose 0.35 plus xylitol 2.06 g/kg body weight. The doses of xylitol were equivalent to human routinely consumed doses (5–10 g/kg body weight). Glucose and glucose plus xylitol solutions were prepared and administered to animals by a repeating pipette (Multipette^®^ plus, Eppendorf, Hamburg, Germany) at 0.5 and 0.1 mL (Combitips^®^, Eppendorf, Hamburg, Germany).

### 2.5. Outcome Measurement

Outcome variables, including the amount of food, water, and hydrogel intake, body weight, adverse symptoms, tumor volume, and survival time, were monitored throughout the experiment. The tumor-bearing mice were examined every other day for the development of tongue tumors and for weight loss [29]. The tumor was measured in two dimensions with a digital 0–100 mm Vernier caliper (Arduino Srl, Genoa, Italy). Tumor volume was calculated as V = AB^2^ (π/6), where A is the longest dimension of the tumor and B is the dimension of the tumor perpendicular to A [29]. The tumor measurements were performed in live animals which are movable; thus, the data from the Vernier caliper could be inconsistent. To ensure the accuracy and consistency in the tumor measurements, a 0.5 cm diameter round-shaped marker was placed above the forceps near the tumor before taking a photograph (Figure 1). The photographs were used for double-checking the accuracy of measured tumor dimensions using ImageJ software version 1.52a (National Institutes of Health, Bethesda, MD, USA).

The mice were sacrificed by CO_2_ inhalation when they had lost >20% of their pre-injection body weight or had become moribund [29]. At necropsy, the tongue tumors and cervical lymph nodes were harvested. The tongue tumor was sectioned in half. One half was fixed in 10% formalin in PBS before histologic studies. The other half was immediately stored in liquid nitrogen before the PFK activity assay.

### 2.6. Histopathological Analysis

The fixed gross specimen was sectioned, processed, and paraffin-embedded. Then, the 5 μm sections were prepared, de-paraffinized, and rehydrated. Hematoxylin and Eosin (H&E) staining was performed according to the manufacturer’s recommendations (Leica Biosystem, Nußloch, Germany). Histologic diagnosis of squamous cell carcinoma and morphological evaluation were performed by an oral and maxillofacial pathologist, who was blinded to the treatment groups. Histologic grading was performed according to the WHO criteria [31].

### 2.7. Immunohistochemical Analyses

The 4 μm sections were prepared, de-paraffinized, and rehydrated. Immunohistochemical staining was performed according to the manufacturer’s recommendations (EnVision kit, DAKO, Agilent, Santa Clara, CA, USA). Briefly, antigen retrieval was performed using citrate buffer, pH 6, for Ki-67 and PFK-1 staining. After endogenous and non-specific blocking, the sections were then incubated with Mouse anti-Human Ki-67 antibody, Clone MIB-1 (DAKO, 1:150 dilution) for 25 min at room temperature or Rabbit anti-Human Muscle Phosphofructokinase/PFKM/PFK-1 polyclonal antibody (Novus Biologicals Inc., Littleton, CO, USA, 1:1000 dilution) at 4 °C overnight. Control sections were incubated with the isotype-matched control antibody. The slides were then visualized using a Dako Envision Flex system.

Tumor cells with Ki-67 nuclear staining were counted and classified into intense (S, G2, M phases) and mild staining (G1 phase), as described previously [32]. Briefly, the photomicrographs of random fields in each slide were taken at the magnification of 400× by a Moticam580INT camera. An oral and maxillofacial pathologist, who was blinded to the treatment groups, counted at least 1000 tumor cells. The Ki-67 labeling index was calculated as the number of Ki-67-positive cells divided by the number of total cells. For the Ki-67-positive cells, the intense/mild staining ratio was calculated as the number of Ki-67 intensely stained cells divided by the number of Ki-67 mildly stained cells.

Tumor cells with PFK-1 cytoplasmic staining were evaluated semi-quantitatively as described previously [33]. Briefly, both intensity (1 = weak, 2 = intermediate, 3 = strong) and percentage of tumor cell staining (0 = less than 10%, 1 = 10–30%, 2 = 31–60%, 3 = more than 60%) were recorded. Then, the combined score, ranging from 0 to 9, was calculated by multiplying both values.

### 2.8. Phosphofructokinase Enzyme Activity

The enzyme activity was measured as described previously [7]. The 6-Phosphofructokinase Activity Assay Kit (Colorimetric) (Catalog #ab155898) was obtained from Abcam (Cambridge, United Kingdom). The frozen tissues were thawed on ice and washed with cold phosphate buffer saline (pH 7.0). Then, the tissues were homogenized with ice-cold assay buffer (containing protease inhibitors) by using a Dounce homogenizer sitting on ice, with 10–15 passes. The homogenates were centrifuged for 5 min at 4 °C at 12,000× *g*. The supernatants were collected for the assay. The protein concentrations of all samples were measured by using the Pierce™ BCA Protein Assay Kit (Catalog #23225). The assay buffer was used to equalize the protein concentrations of all samples. The assay was performed according to the kit’s instructions. The assay is based on sequential reactions of the conversion of fructose-6-phosphate to fructose-diphosphate by PFK, in the presence of ATP, followed by the conversion of ADP from the reaction to AMP and NADH, which converts the colorless probe to a colored product with strong absorbance at OD 450 nm. The activity was measured for 20–60 min at 37 °C in kinetic mode. The rate of PFK activity was calculated as nmol NADH/minute by using a standard curve of NADH at 0–12 nmol.

### 2.9. Statistical Analysis

Comparisons of changes over time in tumor volume, body weight, food, bottled water, and hydrogel intake among groups were analyzed by using two-way analyses of variance (ANOVA), followed by Tukey’s post hoc test. Baseline tumor volumes of each group were compared by one-way ANOVA, followed by Tukey’s post hoc test. Changes in percent survival over time for each group were plotted using the Kaplan–Meier curve, and comparison among groups was performed using log-rank tests. Differences in the Ki-67 labeling index, intense/mild staining ratio, and PFK-1 combined scores were analyzed by using one-way ANOVA, followed by Tukey’s post hoc test. Data analysis and graph generation were conducted with Prism software version 9.3.0 (GraphPad, San Diego, CA, USA). For all comparisons, *p* < 0.05 was considered statistically significant. For survival data, the Bonferroni correction was applied, and *p* < 0.05/3 or *p* < 0.017 was considered significant.

## 3. Results

### 3.1. Establishment of Orthotopic Tongue Cancer Xenograft Model

Thirty out of thirty-five nude mice (85.7%) were found to develop measurable orthotopic CAL-27 tongue cancer xenograft tumors. An example of an orthotopic tongue tumor is depicted in Figure 1.

Histological analysis of tumor-bearing mice exhibited tumor cells with mitosis and characteristic of well-differentiated squamous cell carcinoma (Figure 2A). Moreover, as shown in Figure 2B, perineural invasion (the invasion of tumor cells around the nerves) was observed in tumor specimens of 7 mice (n = 2, 2, and 3 in groups 1–3, respectively). None of the mice showed lymph node metastasis. Histopathological analysis of lymph nodes demonstrated sinus histiocytosis, which is the accumulation of tissue macrophages or dendritic cells. This is a common nonspecific finding in lymph nodes (Figure 2C).

### 3.2. No Significant Differences in Body Weights and No Adverse Effects in Weight Loss of Xylitol-Treated Mice, Compared to the Control Group

The baseline body weight of all animals at tumor initiation (day 1 of treatment) was not significantly different. The average baseline body weights of animals in groups 1–3 were 28.4 ± 1.3, 27.7 ± 2.0, and 28.5 ± 1.5 g/mouse, respectively. Over time, the body weight was reported as a percent change relative to baseline. The average weekly body weight of animals in groups 1–3 on day 7 of treatment were 27.2 ± 3.0, 27.3 ± 2.6, and 28.0 ± 1.5 g/day/mouse, respectively. Body weights of animals in groups 1–3 at day 14 of treatment were 28.2 ± 0.9, 27.2 ± 1.7, and 28.0 ± 1.7 g/day/mouse, at day 21 body weights were 29.0 ± 1.0, 27.2 ± 1.9, and 28.9 ± 2.2 g/day/mouse, and at day 28 body weights were 28.1 ± 3.0, 28.2 ± 1.1, and 28.3 ± 2.7 g/day/mouse. On day 35, body weights of animals in groups 2 and 3 were 28.6 ± 1.7 and 28.6 ± 1.1 g/day/mouse, respectively. As shown in Figure 3, the changes in the average weekly body weights of animals were not significantly different among groups (*p* = 0.787). Interestingly, the maximum percentage of body weight loss in all groups did not exceed 20% of their pre-injection body weight (Figure 3). Such finding indicates no adverse effect from receiving the xylitol.

### 3.3. No Significant Differences in Diet, Water, and Hydrogel Intakes of Xylitol-Treated Mice, Compared to the Control Group

The dietary intake was calculated back from a paste diet to grams of pellet diet. The average dietary intakes of animals in groups 1–3 were 5.9 ± 0.6, 6.0 ± 0.6, and 5.8 ± 0.7 g/day/mouse, respectively (Figure 4A). The average bottled water intakes of animals in groups 1–3 were 1.6 ± 1.2, 1.8 ± 1.3, and 1.5 ± 1.1 g/day/mouse, respectively (Figure 4B). The average hydrogel intakes of animals in groups 1–3 were 3.9 ± 1.2, 3.7 ± 1.1, and 3.6 ± 1.2 g/day/mouse, respectively (Figure 4C). Statistical analyses show that the diet, water, and hydrogel intakes over time were not significantly different.

### 3.4. Non-Significant Reduction in Tumor Volumes of Xylitol-Treated Mice, Compared to Control Group

After tumor initiation, animals were randomly distributed into three groups. The average baseline tumor volumes of animals in groups 1–3 were not significantly different (5.32 ± 4.66, 5.33 ± 3.32, and 5.10 ± 3.07 mm^3^, respectively) (Figure 5A). After the interventions, the tumor volumes in the control group increased over time, while those of the low-xylitol and high-xylitol groups tended to be more retarded (Figure 5B). Nevertheless, the differences were not statistically significant.

### 3.5. Significant Prolongation of Survival in Xylitol-Treated Mice, Compared to Control Group

As shown in Figure 6, the median survival times for control, low-, and high-xylitol groups were 19, 20, and 30.5 days, respectively. The analysis shows that the percent survival of mice given low xylitol and high xylitol was higher than those of the control group. Statistical analyses demonstrated that treatment with a high dose of xylitol significantly prolonged the survival time when compared with the control group (*p* = 0.0105). Such dose is equivalent to 0.1–0.2 g/kg body weight or 10 g/day in humans, which is the household use dose [34].

### 3.6. Significant Decrease in the Intense-to-Mild Staining Ratio of Proliferation Marker Ki-67 in Tongue Tumor of Xylitol-Treated Mice, Compared to Control Group

As shown in Figure 7A–C, the expression of the proliferation marker Ki-67 tended to reduce in xylitol-treated groups. The Ki-67 labeling index in control, low-xylitol, and high-xylitol groups was 0.183 ± 0.124, 0.196 ± 0.079, and 0.116 ± 0.063, respectively (Figure 7D). The index in the high-xylitol-treated group showed a decreasing trend compared to the control group, though it was not significant (Figure 7D). As shown in Figure 7E, the average intense-to-mild staining ratios of the control and experimental groups 1 and 2 were 0.301 ± 0.219, 0.137 ± 0.091, and 0.131 ± 0.086, respectively. The intense-to-mild ratio of Ki-67 staining in the high-xylitol group was significantly lower than that of the control group (*p* = 0.0405). To a lesser extent, such a ratio in the low-xylitol group was borderline significantly less than that of the control group (*p* = 0.0562) (Figure 7E). Mild staining of Ki-67 represents a resting stage of the cell cycle (G0 and G1), while intense staining shows mitotic activity in G2 and M phases. The results suggest that partial substitution of glucose with xylitol shifts the cells towards G0 and G1 phases, with reduced proliferative activity.

### 3.7. Significant Suppression in the Rate-Limiting Glycolytic Enzyme PFK in Tongue Tumors of Xylitol-Treated Mice, Compared to Control Group

As shown in Figure 8A–D, the expression of the rate-limiting glycolytic enzyme PFK was significantly reduced in the high-xylitol-treated group. The PFK-1 combined scores (intensity × percentage of tumor cell staining) in the control, low-xylitol, and high-xylitol groups were 7.36 ± 2.46, 7.11 ± 1.90, and 4.80 ± 2.10, respectively (Figure 8D). The PFK-1 combined score in the high-xylitol group was significantly lower than that of the control group (*p* = 0.0319). Consistently, the PFK activity in the high-xylitol group was lower than that in the control group (Figure 8E), although the difference was not statistically significant (*p* = 0.719) due to high intra-group variations.

## 4. Discussion

Glucose is the primary energy source for normal tissue as well as for cancer proliferation and progression [12]. Understanding the difference in metabolism between cancer and normal cells is an important basis to design appropriate dietary approaches for cancer survivors [6]. Alternative sources of carbohydrates that normal cells can utilize but that do not promote the proliferation of cancer cells are in demand [7,12]. This in vivo study uncovered that partial substitution of glucose with xylitol at the equivalent dose of routine human consumption (10 g/day) significantly prolonged the survival of mice bearing an orthotopic tongue cancer xenograft. Mechanistic studies revealed that xylitol treatment significantly reduced the expression and non-significantly suppressed the activity of the glycolytic enzyme phosphofructokinase (PFK). The reduction in glycolytic activity likely contributes to the significant shift from the cell proliferative stage toward the resting one, as evidenced by the reduced intense-to-mild staining ratio of Ki-67. Such decreased proliferation is consistent with the observed non-significant retardation in tumor growth, as evidenced by lower tumor volumes compared to those of the control group. Interestingly, throughout the experiments, no weight loss exceeding 20% or other adverse events were observed after xylitol treatment. The current study sheds light on xylitol as a candidate sweetener in food products targeting cancer survivors, warranting further clinical studies.

Emerging evidence supports that increased glucose uptake and glycolytic rate are critical for cell proliferation, transformation, and progression of many types of cancer [11,35]. Therefore, targeting glycolytic pathways has been proposed as an approach to cancer therapy [12,14,15]. However, the utilization of this concept for functional food for cancer survivors has never been elucidated. Our previous in vitro study demonstrated that partial substitution of glucose with xylitol (1 mg/mL of glucose and 5.8 mg/mL of xylitol) significantly suppressed the proliferation of oral squamous carcinoma cells (CAL-27, FaDu, and SCC4). In contrast, the growth of non-transformed oral epithelial cells was unaffected. The selective anti-proliferative effect of xylitol likely resulted from an inhibition of glucose utilization through suppression of the rate-limiting PFK enzyme activity and ATP generation [7]. Consistently, our present in vivo study demonstrated that a similar mechanism likely occurs in the animal model of oral cancer xenograft, as evidenced by the significant decrease in PFK expression and the tendency toward reduced PFK activity. Since one gram of xylitol contains lower calories (2.4 kcal) compared to that of glucose (4 kcal), this study was designed to provide partial substitution of glucose with xylitol as an iso-caloric energy input to all groups. Therefore, the findings of this study were likely not due to the deprivation of energy input but rather the lower ATP output generation due to inhibition of glycolysis.

Xylitol utilizes different energy metabolism from that of glucose. Unlike glucose, which can be used to generate ATP by glycolysis, ATP can be created from xylitol using the pentose phosphate pathway [16,36]. Previous studies in cariogenic bacteria have shown that xylitol can suppress phosphofructokinase activity, leading to decreased glycolysis [37,38]. Consistently, our previous in vitro study showed suppression of the activity of the rate-limiting enzyme of glycolysis, PFK, by xylitol treatment. However, our current in vivo study is the first to show that treatment with xylitol can reduce the protein expression of PFK. This result suggests that xylitol may modulate the regulator machinery of PFK synthesis. Previous studies report that gene expression of PFK enzymes is positively regulated by the signal pathway of PI3/K Akt, followed by mTORC1 and transcription factor HIF1-alpha [39]. In contrast, p53 can negatively regulate the glycolytic enzyme synthesis pathway, leading to the slowing down of glycolysis [39]. Therefore, the decrease in PFK expression could be due to either suppression of the Akt-mTORC1-HIF1-alpha pathway or the activation of p53. Since the cell line we used in this study, CAL-27, has been reported to have p53 gain-of-function mutation [29], its glycolytic enzyme expression is expected to be abundant to drive energy generation by glycolysis. Thus, future studies should further explore the molecular mechanism of PFK suppression by xylitol, with special attention to the reactivation of p53 activity.

Since increased glycolysis is an important hallmark in many types of cancer cells [11,12,35], the inhibitory effect of xylitol against glycolysis should have implications for other types of cancer cells. Xylitol has been found to inhibit the proliferation of several types of human cancer cells in a dose-dependent manner, including lung cancer (A549, NCI-H23), kidney cancer (Caki), colorectal cancer (HCT-15), leukemia (HL-60, K562), and malignant melanoma (SK MEL-2) [23]. The previous work addressed xylitol as a possible anti-cancer drug. Nevertheless, based on our knowledge, the current study is the first to explore the potential of xylitol as a functional food for cancer patients. Xylitol is widely used as a sweetener in chewing gum, jam, jelly, baked food, and anti-diabetic food [18]. It has no known toxicity or carcinogenicity in humans and is considered safe [18,40]. Like most sugar alcohol, it has a laxative effect because sugar alcohol is not fully digested in our gut. Consumption of 50 g/day in a liquid drink may cause diarrhea in young adults [41]. A recent review article concluded, based on several human studies, that xylitol up to 30 g is well-tolerated without diarrhea [42], and the average household use is about 5–10 g/day (0.1–0.2 g/kg human/day) [34]. In this study, the household dose was converted to the animal equivalent dose, and partial substitution of glucose with xylitol was explored instead of complete replacement. Such a condition reflects the realistic use of xylitol as a food additive. Since the treatment was applied after the orthotopic tongue tumor was detectable, the finding of this study suggests that partial substitution of glucose with xylitol may be an effective sweetener for oral cancer survivors. Future clinical trials are warranted to confirm this postulation. Owing to the robustly increased glycolysis in many types of cancer, further studies in other cancer models are warranted to confirm its broad applications as a candidate sweetener for cancer survivors in general. Since the Mediterranean diet pattern suggests consuming low sugar, using xylitol as a sugar substitute could be integrated as a part of the Mediterranean diet.

The strength of this study is the orthotopic tongue xenograft model, which is the animal model most resembling human oral cancer. The model was established under the guidance of experts [28,29]. In addition, the intervention was carefully designed with consideration of the household human use dose, partial substitution of sugar as a sweetener, and supply to meet the iso-caloric energy input. Furthermore, this work utilized a texture-modified diet and hydrogel which could help support the nutrition of all animals, allowing adequate time to see the effect of the food-based intervention. Although this study was carefully designed and performed, there were some limitations. First, there was a wide variation in tumor growth among different animals in the same group, resulting in a non-significant decrease in tumor volume and PFK activity. Future studies should use more sample sizes to account for the intra-group variation. Second, the measurement of tumor volume was based on human vision to define the border of the tumor lesion. Therefore, it may not be accurate. A recent study utilized DiR labeling and in vivo imaging to provide better measurement of orthotopic tongue cancer lesions [43]. However, anesthesia is required before performing the in vivo imaging, unlike the Vernier measurement in awake conditions. Future studies should combine both the Vernier measurement for daily monitoring and in vivo imaging for critical periods, e.g., the first detectable time for group assignment, and the period with distinct tumor size among groups. Lastly, besides the inhibition of glycolytic enzymes such as PFK, there could be other mechanisms responsible for the effect of xylitol. For example, a recent study showed that xylitol could target ChaC Glutathione-Specific Gamma-Glutamylcyclotransferase 1 (CHAC1), resulting in decreased GSH levels, inducing oxidative stress, which later leads to apoptotic cell death [44]. More studies are needed to elucidate the tumor-specific mechanism of xylitol.

## 5. Conclusions

Overall, this study showed that partial substitution of glucose with xylitol at the human equivalent dose (10 g/day) retarded tumor proliferation, and prolonged survival of CAL-27 xenograft mice with minimal adverse events, possibly by suppressing the rate-limiting glycolytic enzyme PFK. This study is the first to demonstrate the reduction of PFK protein expression by xylitol treatment and the first to explore xylitol’s potential as a functional food for cancer patients. The promising results warrant future studies in other cancer models and clinical settings. The long-range goals are the broad applications of xylitol as a candidate sweetener in cancer survivors’ novel and medical food products. Furthermore, xylitol could be a candidate sugar substitute to be integrated as a part of the Mediterranean diet to promote cancer survival.

## Figures and Tables

**Figure 1 nutrients-14-02023-f001:**
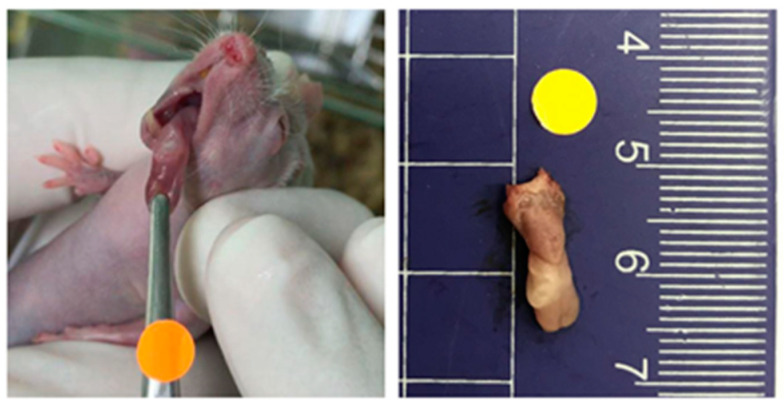
Orthotopic tongue cancer developed by submucosal injection with CAL-27 cells. A round-shaped marker with a 0.5 cm diameter was placed on the forceps when taking tumor photographs.

**Figure 2 nutrients-14-02023-f002:**
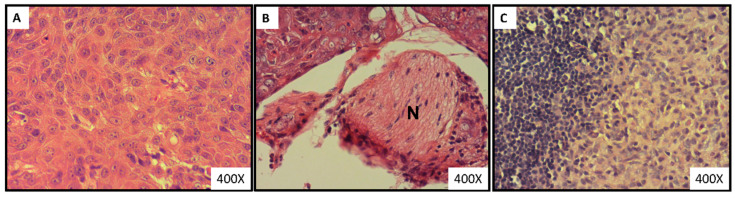
Tongue section of CAL-27 xenograft-bearing mice stained with H&E at 400× magnification. (**A**) Well-differentiated squamous cell carcinomas. (**B**) Perineural invasion surrounding the nerve fiber (N). (**C**) Histiocytosis in the lymph node.

**Figure 3 nutrients-14-02023-f003:**
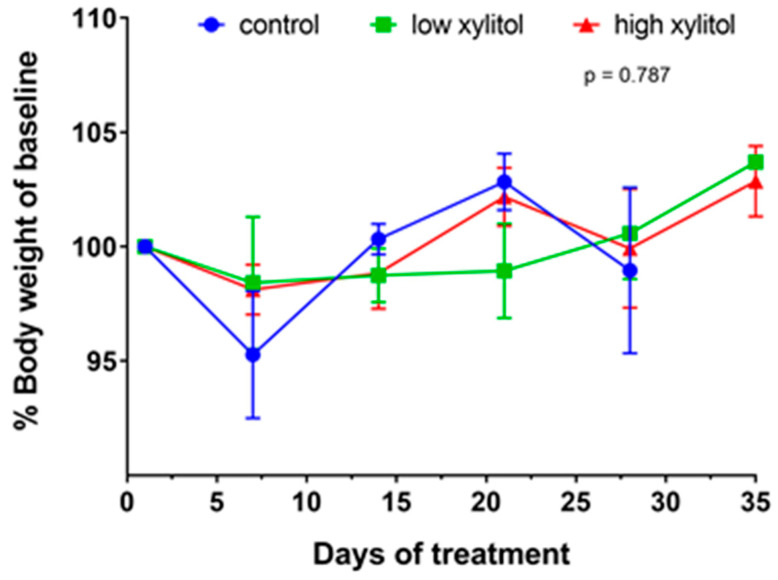
Changes in body weights (percent of baseline body weight) of CAL-27 xenograft-bearing mice receiving glucose solution (control), low-xylitol solution (experimental group 1), and high-xylitol solution (experimental group 2). Results are shown as mean ± SEM. *p*-value was obtained from a two-way ANOVA.

**Figure 4 nutrients-14-02023-f004:**
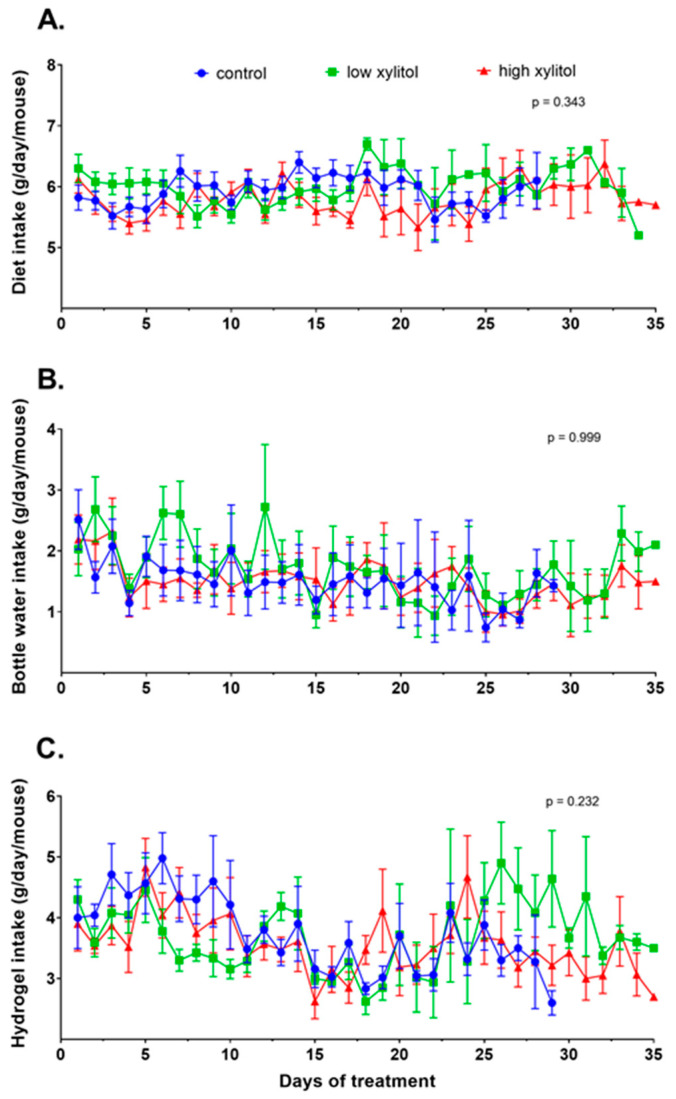
Food and water consumption. (**A**) Average daily diet intake, (**B**) average daily water intake, and (**C**) average daily hydrogel intake of CAL-27 xenograft mice given the glucose solution (control), low-xylitol solution (experimental group 1), and high-xylitol solution (experimental group 2). Results are shown as mean ± SEM. *p*-values were obtained from two-way ANOVA.

**Figure 5 nutrients-14-02023-f005:**
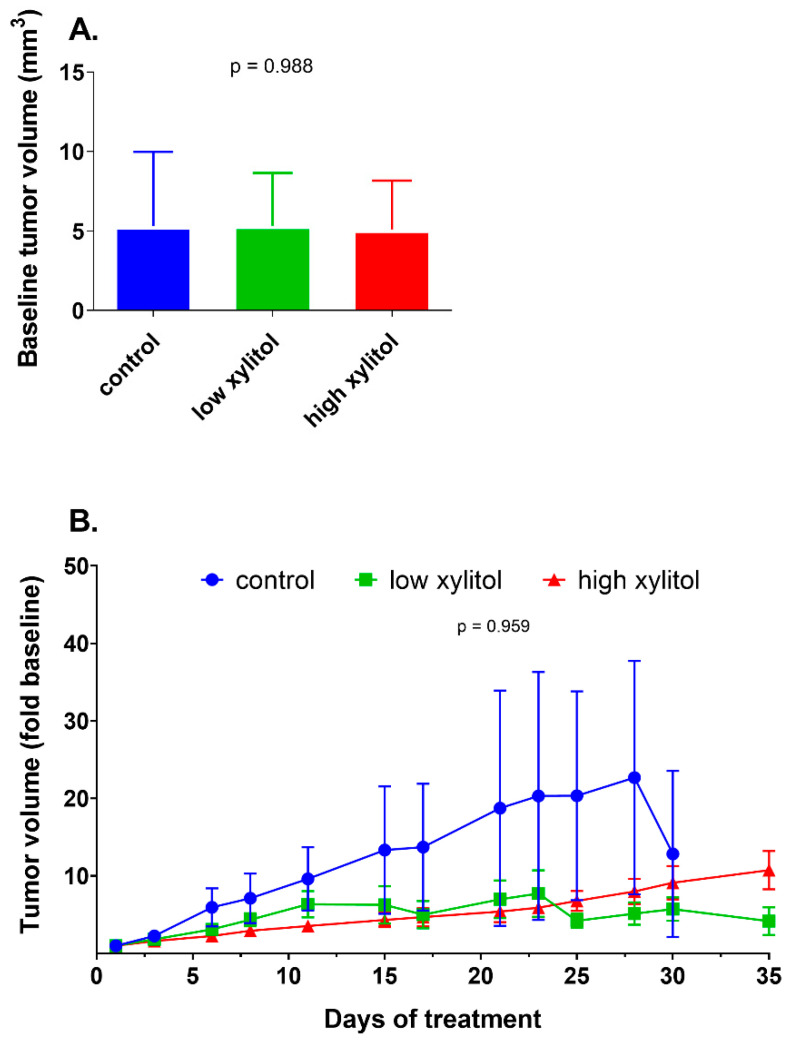
Tumor volumes: (**A**) Baseline tumor volume of CAL-27 xenograft mice given the glucose solution (■ control), low-xylitol solution (■), and high-xylitol solution (■). Results are shown as mean ± SD. *p*-value was obtained from one-way ANOVA. (**B**) Baseline fold change of tumor volume, results are shown as mean ± SEM. *p*-values were obtained from a two-way ANOVA.

**Figure 6 nutrients-14-02023-f006:**
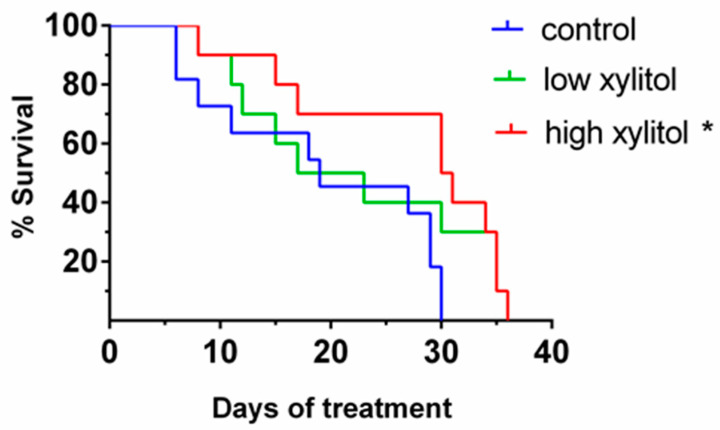
Percent survival of CAL-27 xenograft mice given the glucose solution (control), low-xylitol solution (experimental group 1), and high-xylitol solution (experimental group 2). Changes over time for each group were depicted as Kaplan–Meier curves. * represents *p* < 0.05, obtained from the log-rank test. *p* = 0.0105 comparing between high-xylitol and control groups, *p* = 0.2337 comparing between low-xylitol and control groups.

**Figure 7 nutrients-14-02023-f007:**
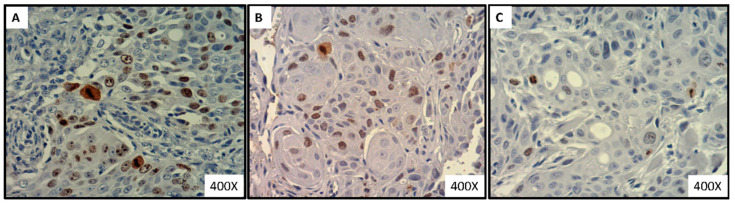
Expression of Ki-67 proliferation marker in tumor sections of CAL-27 xenograft mice. (**A**) Photos were depicted from mice in the control group, (**B**) low-xylitol group, (**C**) and high-xylitol group. (**D**) Ki-67 labeling index of CAL-27 xenograft mice. (**E**) Ki-67 intense-to-mild staining ratios of CAL-27 xenograft mice given the glucose solution (■ control), low-xylitol solution (■), and high-xylitol solution (■). Results are shown as mean ± SEM. * represents *p* < 0.05, obtained by using one-way ANOVA followed by Tukey’s multiple comparison test.

**Figure 8 nutrients-14-02023-f008:**
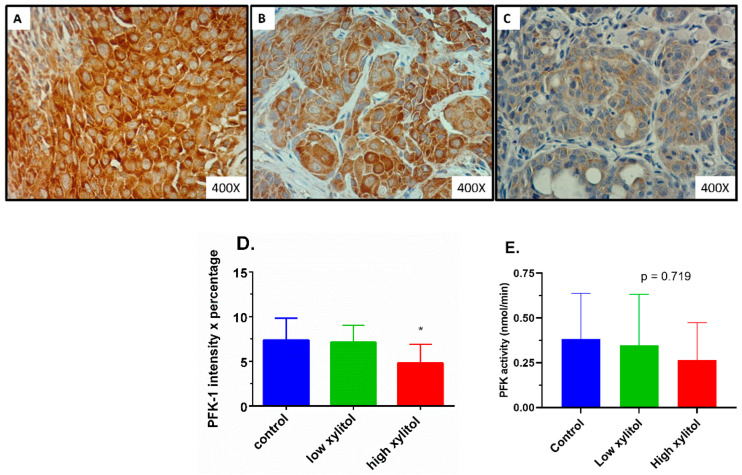
Expression and activity of PFK in tumor specimens of CAL-27 xenograft mice. PFK-1 expression of control (**A**), low-xylitol (**B**), and high-xylitol (**C**) groups. Bar graphs show the mean ± SEM of PFK-1 expression combined score (**D**) or PFK activities (**E**) of CAL-27 xenograft mice given the glucose solution (■ control), low-xylitol solution (■), and high-xylitol solution (■). * represents *p* < 0.05 obtained from one-way ANOVA followed by Tukey’s multiple comparison test.

**Table 1 nutrients-14-02023-t001:** Amount of glucose and xylitol in each experimental group.

Groups	Glucose (g/kg Body Weight)	Xylitol (g/kg Body Weight)	Energy input (kcal/kg Body Weight)
Control group	1.59	-	6.4
Experimental group 1	Low glucose (0.97)	1.03 ^1^	6.4
Experimental group 2	Low glucose (0.35)	2.06 ^1^	6.4

^1^ The doses of xylitol used in animals were calculated from the equation: human equivalent dose (HED) = animal dose × K_m_ human/K_m_ mice. The HED was set at the average household use of 5–10 g/day (0.1–0.2 g/kg body weight). K_m_ human = 37, K_m_ mice = 3.

## Data Availability

No additional data are available.

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
