# Peer review of "Partial Substitution of Glucose with Xylitol Prolongs Survival and Suppresses Cell Proliferation and Glycolysis of Mice Bearing Orthotopic Xenograft of Oral Cancer"

_nutrients, 2022, doi:10.3390/nu14102023_

Round 1

Reviewer 1 Report

The manuscript entitled "Partial Substitution of Glucose with Xylitol Prolongs Survival, Suppresses Cell Proliferation, and Glycolysis in Orthotopic Xenograft Model of Oral Cancer" is an interesting one.
The structure of the manuscript is well organized.
The abstract provides an overview of the entire manuscript, but I would suggest a clear definition of the purpose.
Although for this manuscript I was expecting a more comprehensive introduction, it is short and comprehensive. And in this section I recommend detailing the purpose of this paper.
The material and methods section is sufficiently detailed, and the experiments can be resumed if necessary.
The results are easy to follow, they are clearly presented. The results are interpreted accordingly statistically.
And the discussion section is sufficiently described.
The conclusions section is very short. I would recommend improving this with some conclusions of all the results.

Author Response

  1. The manuscript entitled "Partial Substitution of Glucose with Xylitol Prolongs Survival, Suppresses Cell Proliferation, and Glycolysis in Orthotopic Xenograft Model of Oral Cancer" is an interesting one. The structure of the manuscript is well organized. The abstract provides an overview of the entire manuscript, but I would suggest a clear definition of the purpose.

Response: Thank you for the suggestion. We revised the abstract to clearly define the study objective as “The present study aimed to investigate the effects of xylitol on tumor growth and survival of mice bearing orthotopic xenograft of tongue cancers.”, as shown on page 1 lines 18-19.

  1. Although for this manuscript I was expecting a more comprehensive introduction, it is short and comprehensive. And in this section, I recommend detailing the purpose of this paper.

Response: Thank you for the recommendation. We added more detail on the purpose of the paper in the introduction section on page 2, lines 90-93.

  1. The material and methods section is sufficiently detailed, and the experiments can be resumed if necessary. The results are easy to follow, they are clearly presented. The results are interpreted accordingly statistically. And the discussion section is sufficiently described. The conclusions section is very short. I would recommend improving this with some conclusions from all the results.

Response: Thank you for the recommendation. We revised the conclusion by providing more scientific information, emphasizing the novelty of the research, and adding the explanation and details for potential directions in future research in the conclusion section on page 12, lines 544-553.

Reviewer 2 Report

  1. The title as a statement may be factually misleading.
  2. The 20th century literature items cited in the discussion can be omitted or replaced with modern research.
  3. The methodology, statistical methods, pictures and tables are well described. The discussion is exhaustive.
  4. The conclusions are concrete.
    However, the novelty of the research should be emphasized, and the potential directed in future research (more specifically, details). The conclusions are based on a probable effect - please explain it carefully. Why do the authors suppose so? The conclusions in the abstract are the same as in the conclusions section, so the conclusions section should provide more scientific information.

Author Response

  1. The title as a statement may be factually misleading.

Response: Thank you for your comment. To avoid misleading, we revised the title to “Partial Substitution of Glucose with Xylitol Prolongs Survival, Suppresses Cell Proliferation, and Glycolysis of Mice bearing Orthotopic Xenograft of Oral Cancer.”

  1. The 20th-century literature items cited in the discussion can be omitted or replaced with modern research.

Response: Thank you for the suggestion. We removed all 20th-century pieces of literature and replaced them with modern research papers as shown in the yellow highlight on page 11.

  1. The methodology, statistical methods, pictures, and tables are well described. The discussion is exhaustive.

Response: Thank you for the compliment.

  1. The conclusions are concrete. However, the novelty of the research should be emphasized, and the potential directed in future research (more specifically, details). The conclusions are based on a probable effect - please explain it carefully. Why do the authors suppose so? The conclusions in the abstract are the same as in the conclusions section, so the conclusions section should provide more scientific information.

Response: Thank you for the suggestion. We revised the conclusion by providing more scientific information, emphasizing the novelty of the research, and adding the explanation and details for potential directions in future research in the conclusion section on page 12, lines 544-553.
